# A Global Assessment of Night Lights as an Indicator for Shipping Activity in Anchorage Areas

Semion Polinov [1,2,*] , Revital Bookman [1] and Noam Levin [3,4]

1   Department of Marine Geosciences, Charney School of Marine Sciences, University of Haifa, Mt. Carmel, Haifa 3498838, Israel; rbookman@univ.haifa.ac.il
2   The Chaikin Chair for Geostrategy and Maritime Policy & Strategy Research Center, University of Haifa, Mt. Carmel, Haifa 3498838, Israel
3   Department of Geography, Hebrew University of Jerusalem, Mount Scopus, Jerusalem 9190501, Israel; noamlevin@mail.huji.ac.il
4   Remote Sensing Research Center, School of Earth and Environmental Sciences, University of Queensland, St Lucia, QLD 4072, Australia
*   Correspondence: semion.polinov@gmail.com; Tel.: +972-546569687

**Abstract:** Accurate information on port shipping activities is critical for monitoring global and local traffic flows and assessing the state of development of the maritime industry. Such information is necessary for managers and analysts to make strategic decisions and monitor the maritime industry in achieving management goals. In this study, we used monthly night light (NTL) images of the Suomi National Polar-Orbiting Partnership (Suomi NPP) Visible Infrared Imaging Radiometer Suite (VIIRS) Day/Night Band, between 2012 and 2020, to study the night lights emitted by ships in ports' anchorage areas, as an indicator for shipping activity in anchorage areas and ports. Using a dataset covering 601 anchorage areas from 97 countries, we found a strong correspondence between NTL data and shipping metrics at the country level ($n = 97$), such as container port throughput ($R_s = 0.84$, $p < 0.01$) and maximum cargo carried by ships ($R_s = 0.66$, $p < 0.01$), as well as a strong correlation between the number of anchorage points and the NTL values in anchorage areas across the world ($R_s = 0.69$, $p < 0.01$; $n = 601$). The high correspondence levels of the VIIRS NTL data with various shipping indicators show the potential of using NTL data to analyze the spatio-temporal dynamic changes of the shipping activity in anchorage areas, providing convenient open access and a normalized assessment method for shipping industry parameters that are often lacking.

**Keywords:** VIIRS/DNB; nighttime lights; shipping; spatial analysis; anchorage area





## 1. Introduction

The global rise in the standards of living, consumption volumes, as well as the development and use of marine resources, are leading to an increase in global shipping, despite a temporary slowdown in maritime trade growth in 2018 as a result of trade tensions, protectionism, Brexit [1], and shipping restrictions following the COVID-19 outbreak [2]. According to estimates of the United Nations Conference on Trade and Development (UNCTAD), the volume of international trade by sea accounts for approximately 80% of the volume of world trade [3]. The general trend points to a steadily increasing and developing rate of maritime transport over the past two decades [1]. Ports are important centers of trade between countries, and their cargo handling capacities (loading and unloading of goods) are one of the most basic and important indicators for measuring the development status of ports [4–6]. The port anchorage area (PAA) is a place where ships wait for their turn to enter the port outside the port areas. The PAA is an important part of the shipping and port management segment [5]. Accurate statistical information on the number of ships and cargo loaded and unloaded over a certain period in the port is of decisive importance for monitoring the movement of ships, assessing the state of development of the port, the

country's economy, as well as assessing global trends [6–8]. To date, periodical statistics on port activity are published by various global organizations such as UNCTAD [9], as well as by the ports authorities themselves. Statistical data in frequent comparisons are difficult to access (Lloyd's List Intelligence) [10], varied, and rarely published, making it difficult to analyze data consistently or in near real-time. The recent development of automatic identification systems (AIS), which show the exact location of ships in almost real-time and with an average update of one minute, as well as additional parameters (ship dimensions, etc.) and dynamics (location, direction, speed, etc.), are expensive and complex in processing [11]. While the effects of some anthropogenic activities in marine areas are examined in detail, for example, oil spills from shipping [12,13], our understanding of the spatial and temporal trends of artificial night-time light (NTL) in port anchoring areas remains limited [14].

Conventional methods for monitoring ships at sea from space include optical and synthetic aperture radar (SAR) images obtained using remote sensing. Daytime optical sensors allow the detection of ships; however, their sensors are usually not sensitive enough for detecting low light levels as emitted at night-time [15]. While SAR images have all-weather and day and night capabilities, this approach for detecting ships at sea requires the processing of large amounts of data, and at the moment, there is no operational product offering vessel detection from SAR data [16–18]. Thus, there are still many gaps in the monitoring of ships at daily, monthly and annual time scales. Recent studies have promoted the use of VIIRS low light imaging data for monitoring ships that are using artificial lights within the fisheries industry [19,20]. In this study, we propose the use of night light data to monitor shipping activity in anchorage areas, thereby filling the research gap with a new method for assessing statistical parameters of shipping activities at the port level. This study provides a new approach based on monthly/annual NTL values at different geographic levels (port and country). The challenges we faced included variability in NTL emitted from different ships, the low sensitivity of satellite sensors (VIIRS) to small and weak light sources, and the need for correcting NTL data to minimize the influence of natural light, light from cities, and persistent cloud cover in certain areas [21,22], to measure the magnitude of NTL emitted by ships in the anchorage area.

*Research Question and Objectives*

Our main objective in this paper was, therefore, to determine to what degree can we use NTL as a proxy for shipping activities in anchorage areas, with the following two specific aims:

1.  To what extent can night-time lights in anchorage areas serve as an indicator of shipping activity in port anchorage areas? To do this, we will examine the correspondence between the night-time lights and shipping data at the port level and the country level.
2.  Which variables at the country level can explain the intensity of lighting in anchorage areas? To do this, we will examine various variables that represent economic activities such as GDP, exports, etc. and their correspondence with NTL.

## 2. Materials and Methods

An anchorage area is a place where boats and ships can safely drop anchor. Anchorage areas vary by the types of anchoring which are allowed (size of boat, type of anchor, vessel size, and type) and the authority in charge of the anchorage area (local government, county government, or state government) [23].

### 2.1. Process of Creating the Anchorage Polygons

To examine port activities, we created polygons of anchorage areas using the anchorage points (the offshore location where ships can lower anchors while waiting to be allowed in the ports) dataset of the Global Fishing Watch (GFW) (https://globalfishingwatch.org/data-download/datasets/public-anchorages:v20200316) (v1_20191205) (accessed on 2 August 2020). The anchorage points represent the centers of the anchorage circle (Figure 1). In

different weather conditions, the ship moves radially around the anchor center, so as not to interfere with other ships and the movement of other ships in the port.

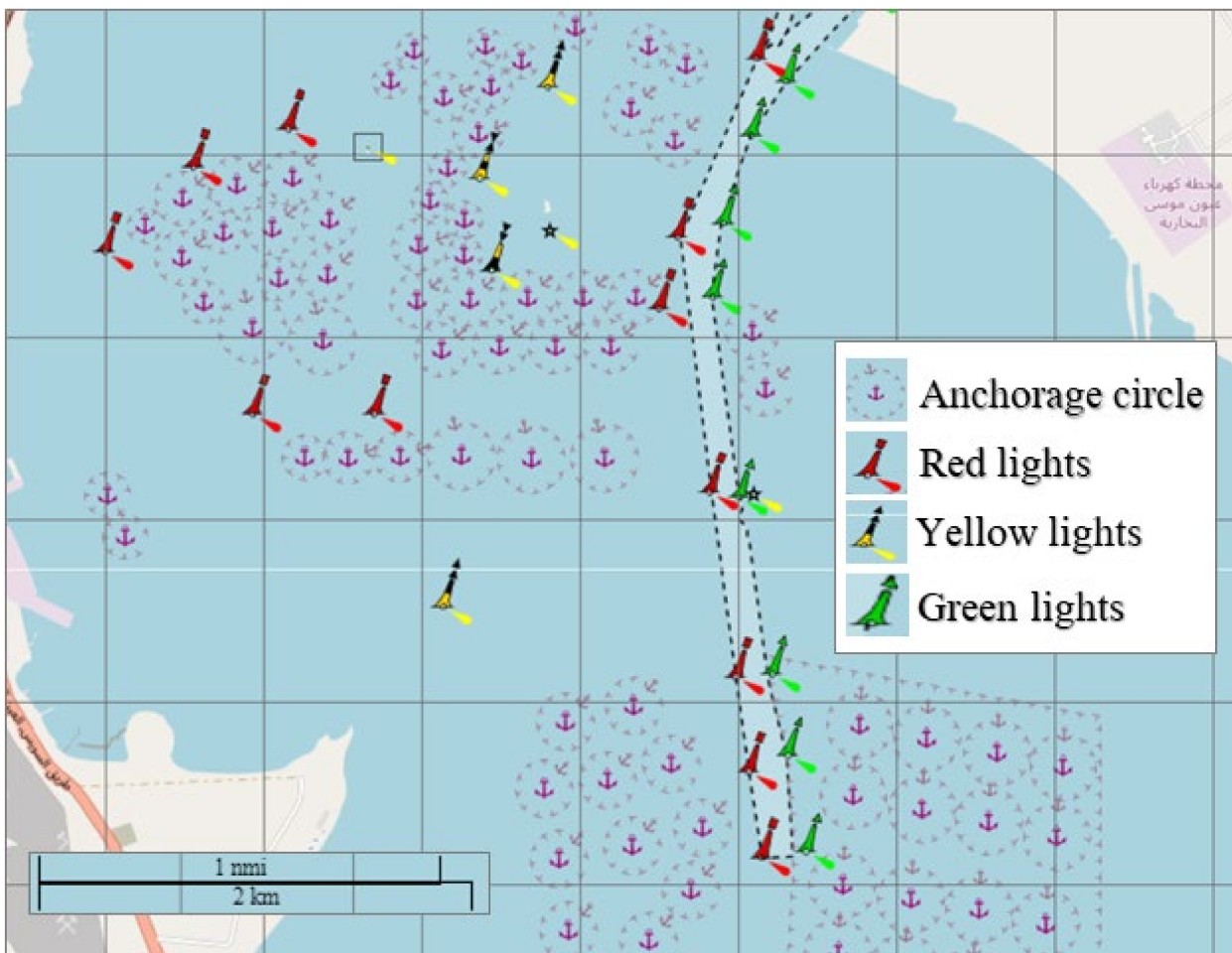

**Figure 1.** An anchorage area at the southern entrance to the Suez Canal. Circles provide anchoring places for different types of vessels, following their size and type (Tanker, Cargo, etc.). In the center of each circle is marked the "anchor", where the ship drops the anchor (source: openseamap.com, (accessed on 10 November 2021)). Colored symbols represent navigation buoys.

To extract the NTL values from VIIRS over the anchorage areas globally, we created a polygonal database of anchorage areas (Figure 2, showing as an example the anchorage areas of Fujairah, United Arab Emirates) covering most of the world's ports. To extract the polygons from the point layer of GFW, first, we transformed the point data to raster format with a cell size of 1 km, using an equal-area projection. We filled the gaps inside each polygon using spatial closing filters with a $7 \times 7$ moving window. We included in our analysis only anchorage areas with at least 10 anchorage points (Figure 2) and which were not close to the coastline so that night-time lights within them will be less affected by coastal urban lights. The night lights emitted by a coastal city can reach offshore areas via scattering by the atmosphere [24]. Moreover, this glow distance varies not only as a result of the density of light sources and the type of lighting but also via atmospheric scattering conditions. The purpose of the following section is to demonstrate the glow of urban coastal night lights into offshore areas.

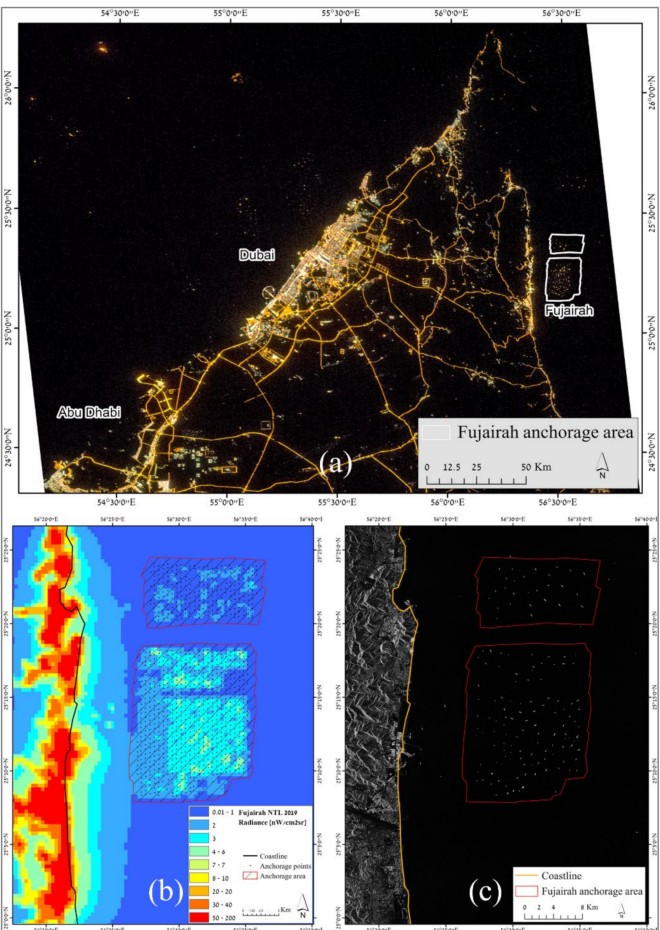

**Figure 2.** Astronaut photography (ISS038–E–16344) of the Eastern part of the Persian Gulf, 2013 (**a**). The average NTL values for 2019 at the vicinity of Fujairah port (**b**), UAE (black dots represents centers of anchorage circles for vessels, based on the dataset of Global Fishing Watch), whereas the (**c**) image shows an image of the same area acquired by Sentinel 1 (9 June 2019). The anchorage area of the port is about ten kilometers from the coastline. Coastal NTL does not affect the anchorage area of the port. The main type of vessel operating in the Fujairah port is tankers.

As can be seen from Figure 2, the urban and coastal areas of Fujairah were much more illuminated by artificial night-time lights (more than 50 nW cm$^{-2}$ sr$^{-1}$) compared with the anchorage area in which the radiance values of night-time lights ranged between 2–7 nW cm$^{-2}$ sr$^{-1}$. Despite the strong night illumination of the city, it did not affect the amount of light in the area of the anchorage area since the influence of city lights from Fujairah ended about 6–7 km from the coastline.

The resulting anchorage polygons which we created included 601 anchorage polygons belonging to 97 countries. Finally, these anchorage areas included 44,570 anchorage points out of a total of 119,478 anchorage points in the original dataset of GFW.

### 2.2. VIIRS Night–Time Light Data (Response Variables)

The VIIRS DNB data are more sensitive to low light levels than the DMSP/OLS and have a higher spatial resolution of 742 m × 742 m footprint from nadir out to the edge of scan [25]. The monthly VIIRS products are gridded to a 15 arc-second grid, which is slightly finer than the original pixel footprints [26]. In this study, we used VIIRS DNB monthly cloud–free average data products for the period between April 2012 and March 2020 provided by the NOAA service (https://ngdc.noaa.gov/eog/viirs/download_dnb_composites.html, (accessed on 20 June 2021)). The dataset includes radiance data and cloud-free coverages (the number of cloud-free acquisitions available for a given month

for calculating night-time brightness in each pixel). The DNB radiance data excluded data impacted by stray light, lightning, lunar illumination, and cloud cover before averaging, while some temporal lights from auroras, fires, boats, etc., are reserved [27]. Two configurations of the VIIRS composites are available: "vcmcfg" excludes any data contaminated by stray light (typically solar illumination) and "vcmslcfg" excludes data impacted by stray light are corrected but not removed. We selected the "vcmslcfg" products as they offer greater temporal and spatial coverage [28]. Moreover, we calculated the annual sum of lights within each anchorage area by multiplying the polygon area with the average values of NTL. We used the equal–area Mollweide projection to calculate anchorage areas and to calculate the sum of lights (SOL) of anchorage areas. To extract NTL values of anchorage areas data, we used the Google Earth Engine (GEE) platform [29]. We extracted and merged both data configurations: 'VCMSLCFG' for the period of January 2014–March 2020 and 'VCMCFG' for the period of April 2012–December 2013, of two available bands:

- "Avg_rad"—value represents the monthly average value of NTL.
- "Cf_cfg"—cloud–free days (this was important to interpolate monthly radiance values for months that were too cloudy, as detailed below).

We used the correction coefficients provided by Coestfield [22] for correcting the temporal variation of natural light sources such as airglow and thus corrected the monthly time series of NTL using their published coefficients, which are especially important in areas with low levels of NTL.

In various anchorage areas, NTL values were underestimated in certain months due to high cloud cover for most of the month. We used a threshold of an average of at least one day without cloud cover per month within the anchorage area as the minimum threshold in which we accepted NTL values for a specific month that would be "valid" for our analysis of the anchorage areas. According to our dataset of NTL for 96 months and 601 anchors, almost 95% of the months in all ports had sufficient cloud–free data (Table 1). In the majority of cases, cloudy months (with cloud–free days below 1) were usually isolated (3.1% of all months; Table 1). More than 50% of all anchorage areas experienced at least one event of two consecutive months with persistent cloud coverage (cloud–free days below 1; Table 1).

**Table 1.** Distribution of the number of months without sufficient NTL data measurements due to high cloud cover. The left–hand column shows the number of consecutive months without sufficient cloud–free measurements (average cloud–free days < 1). Our dataset included 96 months and 601 anchorage areas (96 × 601 = 57,696). The first group, "0 = Zero", indicates months with no missing data.

| Number of Consecutive Missing Months with Cloud Free Days (CFD) Value < 1 | Sum of Months in Each Group across all Anchorage Areas | % of the Sum of Months in Each Group across all Anchorage Areas (out of 57,696) | Number of Anchorage Areas in Each Group of Missing Months | % Number of Anchorage Areas in Each Group of Missing Months (out of 601) |
|---|---|---|---|---|
| 0 | 54,589 | 94.6% | 113 | 18.8% |
| 1 | 1772 | 3.1% | 488 | 81.2% |
| 2 | 731 | 1.3% | 344 | 57.2% |
| 3 | 375 | 0.6% | 235 | 39.1% |
| 4 | 149 | 0.3% | 126 | 21.0% |
| 5 | 56 | 0.1% | 53 | 8.8% |
| 6 | 24 | 0.04% | 24 | 4.0% |

To fill the monthly gaps in NTL values, we interpolated NTL values for months with average cloud–free days of less than one day, based on the six months before and after the missing month. Based on this method, the missing values were filled in, and the original values for months with CFD $\geq$ 1 were not changed (see example in Figure 3 for the anchorage area of Punta Arenas, Chile).

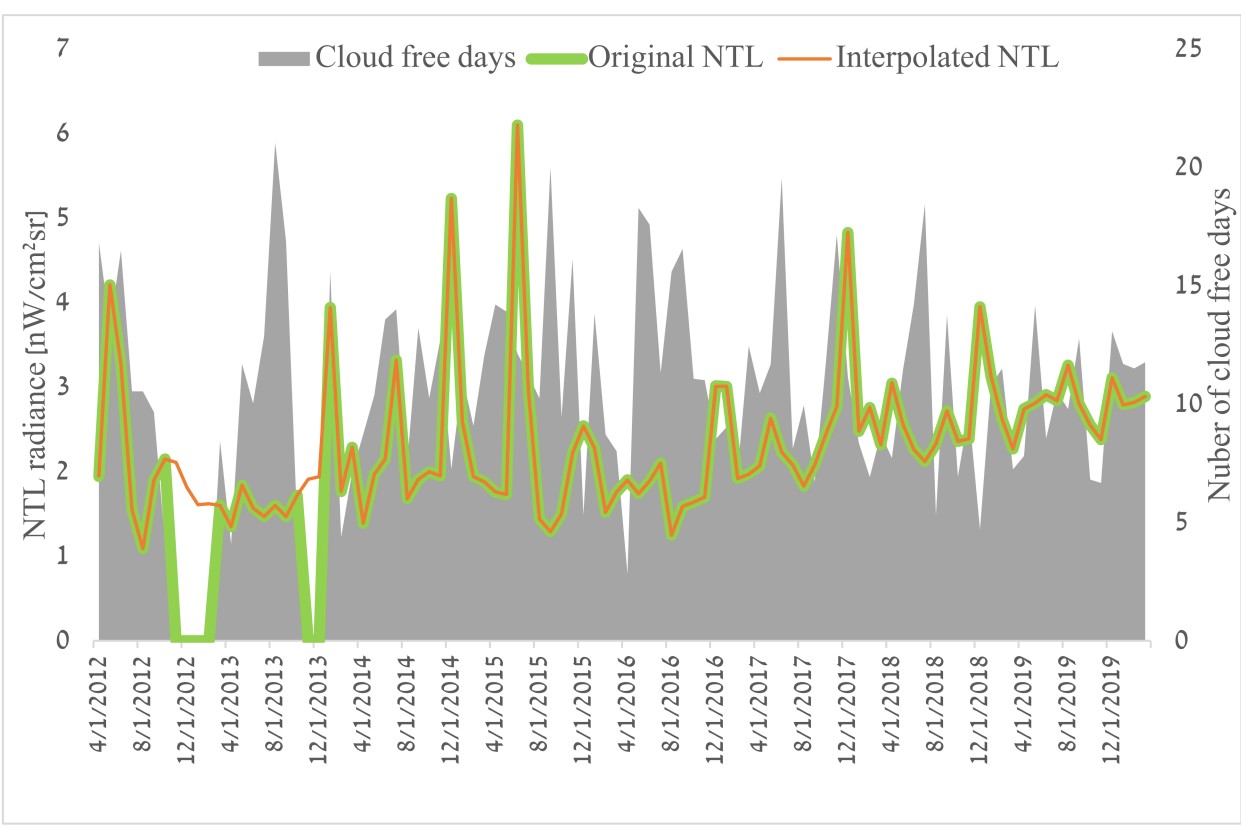

**Figure 3.** Interpolation of NTL values for cloudy months (for the anchorage area of Punta Arenas, Chile).

Finally, to estimate the impact of the interpolation method of "Interpolating NTL for cloudy months" on the average NTL values in all anchorage areas, we performed a 2-tails *t*-test. The test was used to determine if there was a significant difference between the means of the original NTL values vs. the interpolated values for each of the anchorage areas. Running a *t*-test between the NTL time series of each anchorage before and after the interpolation of months in which the number of the cloud–free day was less than one, we found that the null hypothesis ($H_0$—no difference between the original monthly values and the interpolated ones) could not be rejected (see example Figure 3 for Punta Arenas anchorage area) for any of the anchorage areas (i.e., the average NTL values before and after the interpolation were not different).

### 2.3. Explanatory Variables

We collected a range of quantitative information on maritime activities at the port level, as well as various parameters (economic, environmental, etc.) at the country level, using data sourced from the United Nations Conference on Trade and Development (UNCTAD) (https://unctad.org/statistics, accessed on 12 October 2021), the World Bank (https://data.worldbank.org, accessed on 12 October 2021), and www.trademap.com (accessed on 12 October 2021) at the port and country–level (Table 2). The UNCTAD "Port calls/Port performance" parameters group is one of the main sources of maritime shipping data and is part of a suite of port call and performance statistics that provide an overview of the characteristics of ships and the time they spent in a country's ports over a given period [30,31]. Together, the "Port calls/Port performance" includes statistics for up to eight parameters, covering: Median time in port—the median time vessels spent within port limits (in days); Average size of the vessel—the average gross tonnage of the vessels that have called in the country's ports during the year, as well as the "Maximum" parameters of the largest ships that have called during the period, the maximum cargo carrying capacity and the maximum container carrying capacity of container ships; Average cargo carrying

capacity—the average deadweight tonnage of the vessels that have called in the country's ports during the year; Average container carrying capacity per container ship—the average capacity to carry the 20–foot equivalent units (TEU) of the container ships. Another data source from the UNCTAD is the container port throughput (CPT)—measurements of the container flow from land to sea transport modes, and vice versa, in 20–foot equivalent units (TEUs), a standard–size container [32]. Data refer to coastal shipping as well as international shipping. Trans–shipment traffic is counted as two lifts at the intermediate port (once to off–load and again as an outbound lift) and includes empty units. The liner shipping connectivity index (LSCI) represents the country's integration level into global liner shipping networks [33,34]. The LSCI is an index set at 100 for the country with the maximum value of country/port connectivity in the first quarter (Q1) as of 2006, which was then China. From the Wordlbank, we have downloaded the following data at the state level: Electric power consumption—electric power consumption per capita (kWh), which are the main indicators of the size and level of development of the country's economy; Fossil fuel consumption (% of total)—comprises coal, oil, petroleum, and natural gas products by country; Population—a country population based on national population censuses. Most of the explanatory variables used in the study represent the average values over a certain period (Table 2). Since the NTL data we used covered a wider time range (2012–2020), we calculated, for each of the explanatory variables, the mean over the period corresponding to the response variable (NTL data) or the widest time range that could be extracted, for example, TSC data were available for 2016–2020. Most of the explanatory variables were only available at the country level and only a few at the port level. These data describe various aspects directly or indirectly related to maritime ship activity in the ports at the country level and thus may explain the number of ships in the anchorage areas.

**Table 2.** List of explanatory (independent) variables at the country and port level.

| Parameter | C—Country P—port | Years | Data Source |
|---|---|---|---|
| Number of anchorage points | C/P | 2019 | [35] |
| Average cargo carrying capacity | C | 2018–2020 | [31] |
| Average container carrying capacity | C | 2018–2020 | [31] |
| Average size of vessel | C | 2018–2020 | [31] |
| Average $CO_2$ emissions | C | 2016 | [36] |
| Container port throughput (CPT) | C | 2016–2019 | [37] |
| Electric power consumption | C | 2013–2014 | [38] |
| Fossil fuel consumption | C | 2013–2015 | [39] |
| Gross domestic product (GDP) | C | 2016–2020 | [40] |
| GDP growth (annual %) | C | 2016–2020 | [41] |
| Import | C | 2016–2020 | [42] |
| LSCI | C | 2016–2020 | [31] |
| Maximum cargo carrying capacity of vessels | C | 2018–2020 | [31] |
| Maximum container carrying capacity of vessel | C | 2018–2020 | [31] |
| Maximum size of vessels | C | 2018–2020 | [31] |
| Median time in port (days) | C | 2018–2020 | [31] |
| Population growth (%) | C | 2016–2020 | [43] |
| Population total | C | 2016–2020 | [44] |
| Monthly average number of vessels in the PAA of Santos | P | 2016–2020 | Sentinel 1 |
| Santos port statistics (Import/Export, ship numbers by class and by waiting time) | P | 2016–2020 | [45] |



Data collection on Vessel Numbers in Anchorage Areas from Sentinel 1

To further examine the potential of using NTL as a measure of temporal changes in ship activity in ports, we collected data on the number of ships anchored in the port anchorage area. Using Sentinel 1 (SAR) satellite images (which offer all–weather capabilities and enable ship detection; [46]) from January 2016 to March 2020 in the port of Santos (Brazil's largest port), the average number of vessels sighted in the anchorage area was calculated based on 4–5 images per month. The port of Santos was chosen because it is a relatively large port with a remote anchorage area. Such a large anchorage area can accommodate more than 50 ships at any given time, which can create a large amount of night–time light that will not be affected by city lights. Thus, a port such as Santos is a good place to test the hypothesis that night lights may serve as an indicator of temporal changes in shipping activity in the anchorage area.

To perform a detailed analysis at the port level for Brazil's largest port (the port of Santos), the following statistical data from the port's website (http://www.portodesantos. com.br/informacoes-operacionais/estatisticas/mensario-estatistico/ (accessed on 1 February 2022)) were also used: monthly volumes of imports and exports, the monthly number of ships waiting (total and those waiting >72 h) in the anchorage area, and the number of ships from different segments (general cargo, bulk solids, bulk liquids [tankers], passengers, etc.) that visited the port of Santos.

*2.4. Analysis*

We conducted a correlation analysis at two levels: at the port level and the country level. We chose to include a country–level analysis as well, as it is useful in our view to conduct between–countries comparisons, which is also common practice in other studies of economic activities using night lights [28,47,48]. For each spatial level, various explanatory and response variables were prepared (average NTL and Sum of Lights). For the country–level analysis, we calculated the mean values of the explanatory parameters and the mean value of the NTL of all anchorage areas of that country. At the port level, a comparison was made between the monthly average NTL and the average number of vessels, using the example of the port of Santos, Brazil. Spearman's rank correlation coefficient (denoted here by Rs) was calculated using SPSS to examine the correspondence between the average annual NTL 2012–2020 and the explanatory variables (Table 2). The use of NTL was proven to be effective for the study of large areas, such as at the country, state, county, or city level [49].

**3. Results**

Overall, we identified and analyzed 601 anchorage areas representing 97 countries with a temporal coverage of 96 months (April 2012–March 2020). In this study, an assessment was made of the level of the correlation between the average values of VIIRS (monthly, annual) sum of lights (SOL) both at the level of a single anchorage and at the country level.

The map of the distribution of the average annual NTL value for 2012–2020 (Figure 4) indicates high concentrations of anchorage points per port along the coasts of China (Figure 4c), the Persian Gulf (Figure 4b), the Mediterranean Sea (Figure 4a), Gulf of Guinea, the Southern Coast of Brazil, and the Southern Caribbean, which indicates high shipping activity in these regions during the study period. The spatial distribution of ports by their number of anchorage points varies greatly, with the highest number of anchorage points found along the coast of China. The anchorage area of the port of Xingang (China) had the highest number of anchorage points (1197), followed by the port of Fujairah (United Arab Emirates) with 725 anchorage points. The port of Shanghai had several anchorage areas, which combined included more than 1300 anchorage points; The port of Lome (Togo), with 620 anchorages, was the largest of any African port. Of the European ports, Malta had the largest number of anchorage points (498). The Brazilian port of Santos was the leading port in the number of anchorage points (449) within the Americas.

Figure 5 show a map of the anchorage area and their average annual sum of lights (SOL) values for 2012–2020. On overage, the ports of the Persian Gulf (Figure 5b) were among the most brightly lit (112 nW/cm$^2$cr). Anchorage areas located in Asia (especially in the East China Sea, Figure 5c) had significantly lower average values (43 nW/cm$^2$cr). The anchorage areas of the American continent, especially the northern part, were much less lit, with an average of 39 nW/cm$^2$cr. Of the seas of the European continent, the anchorage areas of the Mediterranean Sea (Figure 5a) were the most lit with an average of 48 nW/cm$^2$cr, with Malta's being the most lit with 443 nW/cm$^2$cr. The average NTL values of the anchorage areas of European ports in the Atlantic Ocean and the Northern Sea had lower SOL values (23 nW/cm$^2$cr), while the most lit anchorage area (Rotterdam) was ranked globally as 23rd most lit anchorage area with an average SOL of 187 nW/cm$^2$cr. China's anchorage areas had relatively low SOL values (average of 45 nW/cm$^2$cr based on 87 areas, with only 10 anchorage areas with SOL values higher than 100 nW/cm$^2$cr), despite a high average number of anchorage points of 155 per anchorage area (*n* = 87). The anchorage areas of Xingang and Ningbo (China), with an average of 354 and 328 nW/cm$^2$cr, respectively, were the most lit among Chinese ports regarding anchorage areas. Anchorage areas along the East Coast of Africa were less lit (37 nW/cm$^2$cr) than those of the West Coast of Africa (59 nW/cm$^2$cr) and the Southern Coast of the Mediterranean Sea (57 nW/cm$^2$cr). The port areas of Luanda (Angola) and Lome (Togo) were the most lit on the African continent, 252 nW/cm$^2$cr and 240 nW/cm$^2$cr, respectively. The anchorage area of the port of Kandla was the most lit (523 nW/cm$^2$cr) among all the investigated anchorage areas in India. The Port of Kandla had the largest number of anchorage points among all ports in India (178 anchorage points) and was ranked 53rd out of all anchorage areas in terms of its number of anchorage points.

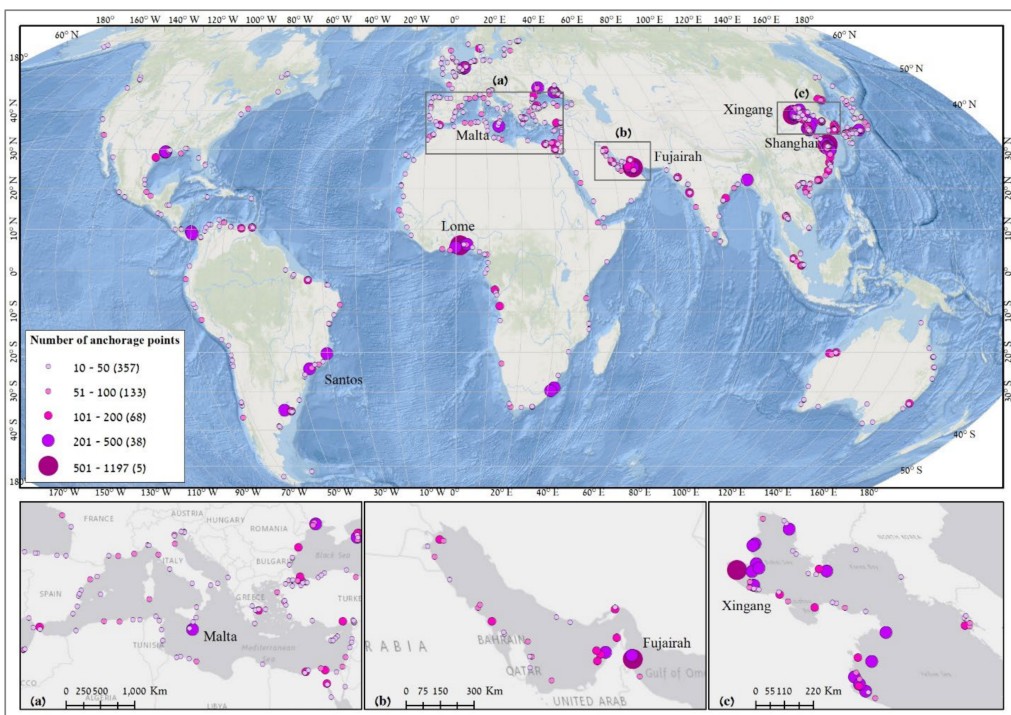

**Figure 4.** Map of anchorage areas (ports) symbolized by their number of anchorage points (source of anchorage points: www.globalfishingwatch.org, accessed on 5 October 2020). Enlarged maps show the following areas: (**a**) The Mediterranean Sea; (**b**) The Persian Gulf; (**c**) The Yellow Sea region.

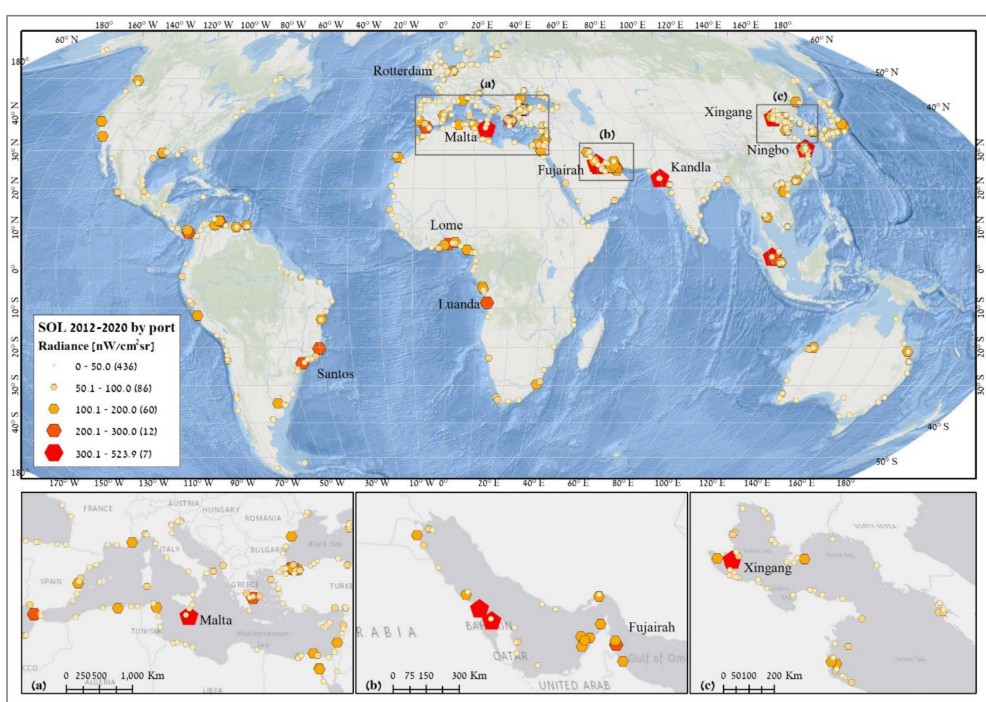

**Figure 5.** Map of anchorage areas and their average annual sum of lights (SOL) intensity for the period of April 2012–March 2020. Enlarged maps show the following areas: (**a**) The Mediterranean Sea; (**b**) The Persian Gulf; (**c**) The Yellow Sea region.

### 3.1. General Patterns of NTL as an Indicator of Shipping Activity

In this section, we present the results of the correlation analysis obtained at the global level. The purpose of this section is to assess the correlation between two fundamental parameters, NTL and the number of anchorage points at different geographical levels

Figure 6 show a statistically strong and significant ($R_s = 0.69$, $p < 0.01$) correlation between the number of anchorage points per area and the average annual value of NTL sum of light (SOL) for 2012–2020 based on 601 anchorage zones. This result indicates a strong relationship between the total number of anchorage points for ships within the anchorage area and the average SOL reflected from anchorages at sea. A higher number of anchorage points increases the amount of NTL lights measured by the satellite sensors.

Figure 7 present the correlation between the total number of anchorage points in the ports of each country (97 in total) and the average NTL value for 2012–2020. Despite the similarity of the two graphs (6 and 7), a higher correlation was obtained at the country level ($R_s = 0.84$, $p < 0.01$) between anchorage points and NTL than at the port level. In countries with a developed maritime industry such as China, Japan, Turkey and the USA, a large number of anchorage areas exist, and as a result, they emit high levels of NTL.

Figure 8 show a scatter plot of the correlation analysis between the annual average SOL and the signal to noise ratio (SNR = Average/Standard Deviation) for the period 2012–2020. We obtained a significantly strong correlation ($R_s = 0.49$, $p < 0.01$) between the annual average SOL value and the SNR over the investigated period of 2012–2020. Anchorage areas with SNR values below one indicate high variability in monthly NTL values. Out of 601 anchorage areas, 33 (5%) had SNR values below 1, while China had 12 (13% of all Chinese anchorage areas) and Germany had 5 and 7 anchorage areas. Anchorage areas with SNR values above 2 represent are areas with more stable monthly/annual NTL. The most stable NTL values were obtained in the anchorage areas of the port of Tenerife, Spain (Average/Stdv 258.37/19.9 nW/cm$^2$sr, SNR = 13), Valencia, Spain (166.45/13.7 nW/cm$^2$sr, SNR = 12.2), and Kuwait (164.56/15.5 nW/cm$^2$sr, SNR = 10.6), while the most unstable of the highly active ports (with average NTL above 150 nW/cm$^2$sr) was the anchorage area of Khalifa Bin Salman, Bahrain (153.9/484.8 nW/cm$^2$sr, SNR = 0.3).

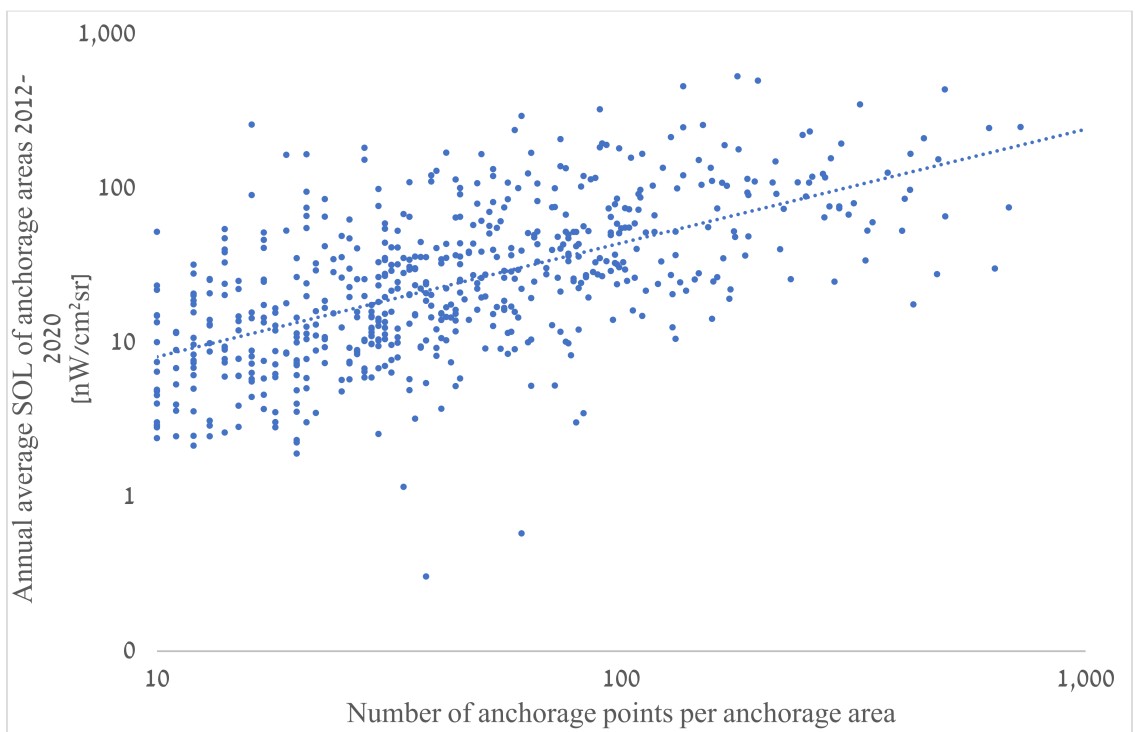

**Figure 6.** Spearmen correlation between the number of anchorage points per anchorage area and averages of annual SOL from 2012 to 2020 ($R_s = 0.69$, $p < 0.01$), N = 601.

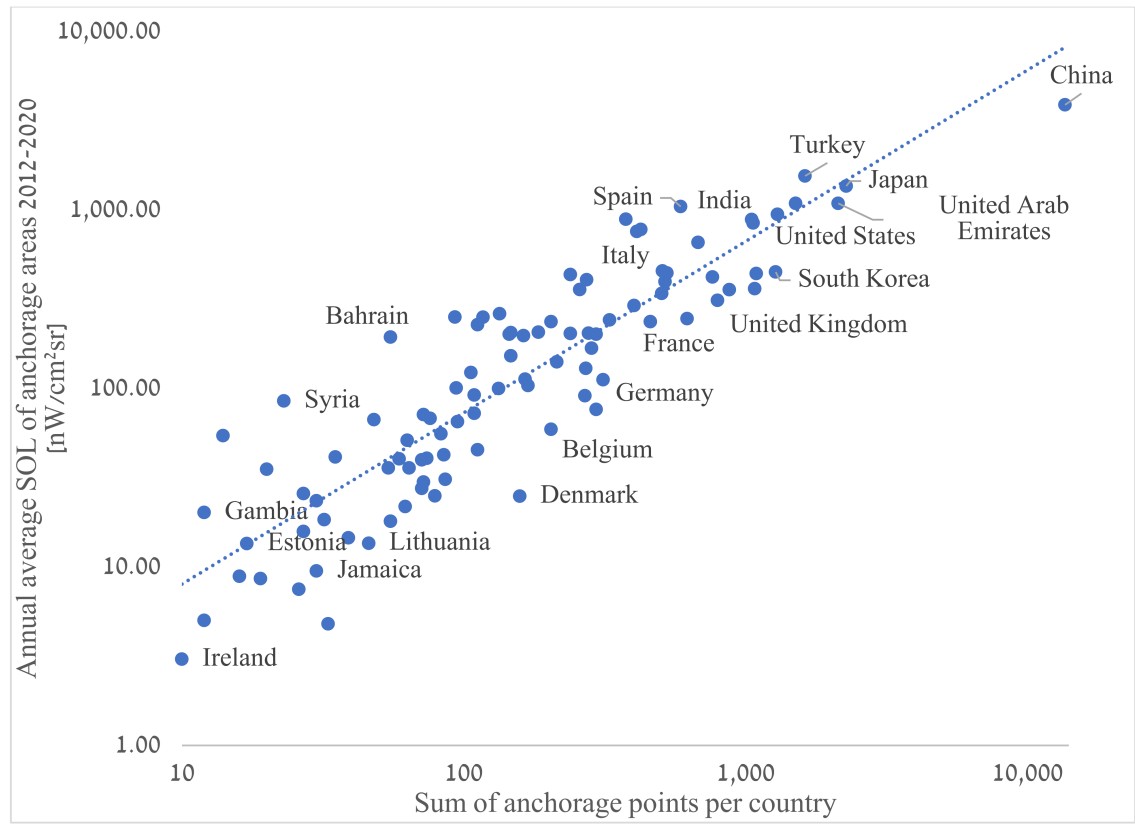

**Figure 7.** Logarithmic scatter plot of Spearmen correlation between the total number of anchorage points by country and averages of annual NTL sums, from 2012 to 2020 ($R_s = 0.84$, $p < 0.01$), based on N = 97.

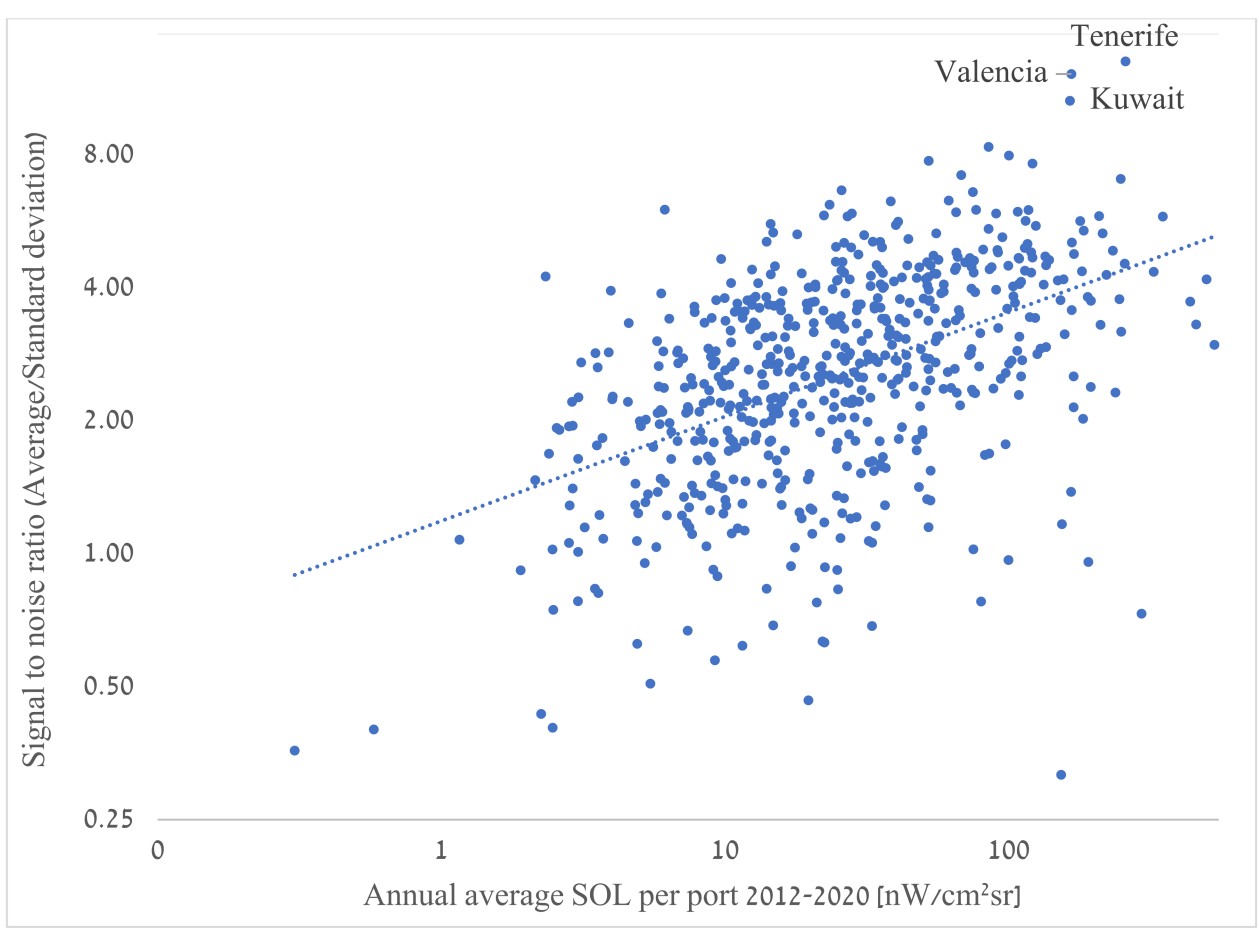

**Figure 8.** Scatter plot of all anchorage areas' average SOL and the signal to noise ratio (SNR = Average/Standard deviation) for the period of 2012–2020. ($R_s = 0.49$, $p < 0.01$), based on N = 601.

### 3.2. Temporal Analysis of NTL Values within Anchorage Areas

Figure 9 and Table 3 present the results of a correlation analysis between the NTL values within the port of Santos, Brazil and different statistical parameters of the port and the number of ships counted from Sentinel 1 images. The purpose of this section is to assess the feasibility of using the monthly average of NTL values as an indicator of the port shipping activity by estimating the number of ships in an in anchorage area on a monthly basis. The port of Santos is a relatively large port with a large anchorage area in which, during the study period, the average daily number of ships in the anchorage area was 60 with a standard deviation of 16 ships per month. The average monthly number of ships waiting for more than 72 h was 110 with a standard deviation of 23.

Over most of the study period, the five parameters in Figure 9 show statistically significant correlations, with corresponding peaks and lows. In the first half of 2019, the NTL values had a local peak which was not present in the number of ships as counted from Sentinel 1; however, this peak was found in the variables of total exports and ships waiting for more than 72 h.

VIIRS monthly sum of light values were moderately correlated with the number of ships counted from Sentinel 1 images ($R_s = 0.51$), the number of ships carrying bulk solids ($R_s = 0.41$), and the number of ships that waited for more than 72 h in the anchorage area ($R_s = 0.41$) (Table 3). Strong correlations (Rs $\geq$ 0.72; Table 3) were found between all combinations of the following pairs of variables: the number of ships waiting, the number of ships waiting for more than 72 h (being on average 29% of all ships waiting), bulk solids ships (being on average 26% of all ships), and exports. Moreover, the number of bulk solids

ships correlated with several other parameters: ships counted from Sentinel 1 ($R_s = 0.5$) as well as with exports ($R_s = 0.87$), while for the imports, a weaker correlation was found ($R_s = 0.39$). We also observed good correlations between the total monthly number of ships (based on the port's official statistics) and monthly exports ($R_s = 0.74$), the monthly number of bulk solid ships ($R_s = 0.72$), the number of ships counted from Sentinel 1 ($R_s = 0.48$), and with the monthly number of general cargo ships ($R_s = 0.47$) (Table 3).

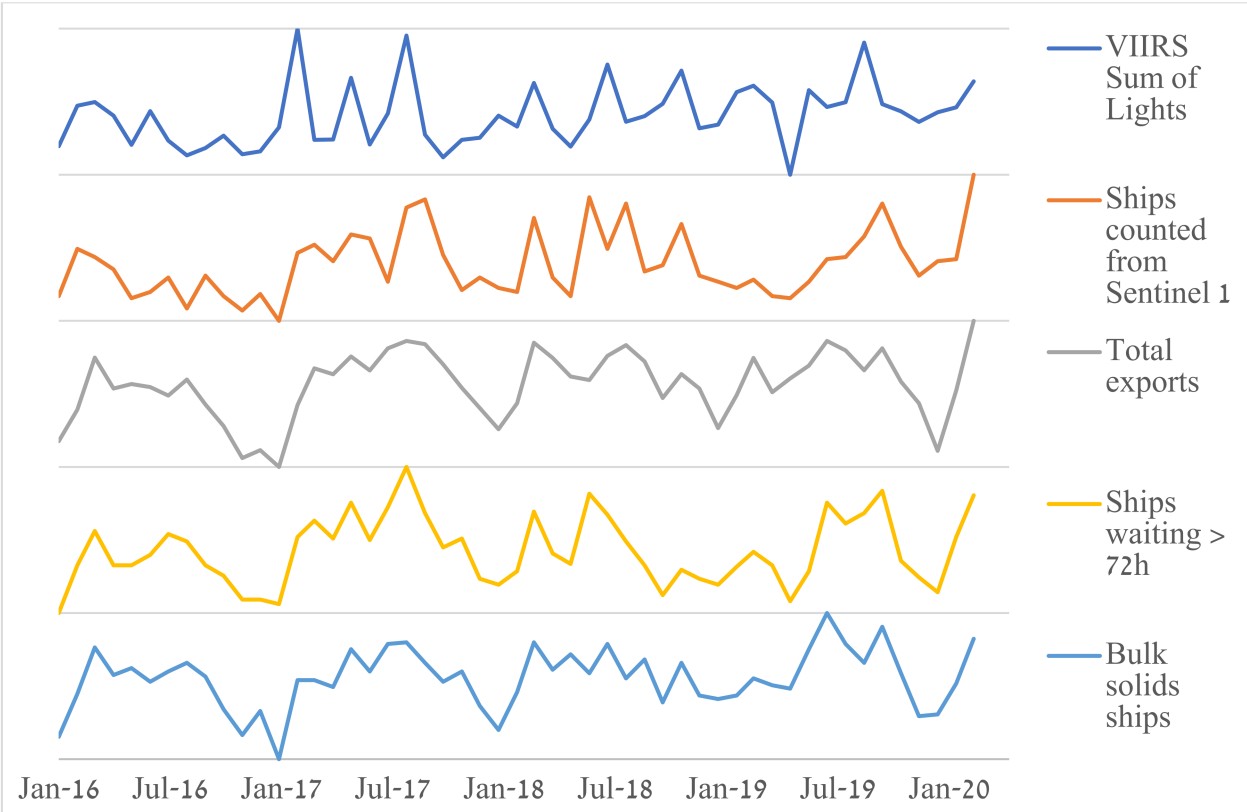

**Figure 9.** Time series presenting the correspondence between the average number of vessels counted from Sentinel 1, total exported goods, number of vessels waiting for more than 72 h in anchorage area, number of ships carrying bulk solids and average NTL over the anchorage area of Santos port, Brazil. All variables were normalized between their respective minimum and maximum values from Jan 2016 to Mar 2020 to ease the visual comparison between them.

Extending the temporal analysis to all anchorage areas, Figure 10 show a global map of temporal trends in NTL within anchorage areas for the period 2012–2020. In 219 (36%) of the anchorage areas, temporal trends were not significant (*p*-value > 0.05), with a relatively even distribution across continents. Of the 382 anchorage areas in which we found statistically significant results, in 112 (18%) of the anchorage areas, there was a decrease in NTL, while in 270 of the anchorage areas, there was an increase in NTL values. In the Mediterranean region (Figure 10a), the temporal trend of the NTL value of anchorage areas showed an increase (avg. $R_s = 0.31$, $p < 0.01$, Stdv = 0.38) based on 67 observations, while only 13 anchorage areas recorded a decrease. The NTL values growth in the Mediterranean region was mainly due to Turkey's anchorage areas, where 12 anchorage areas had a statistically significant and strong temporary trend of increase (avg. $R_s = 0.6$, $p < 0.01$). On the African continent, a significant difference in temporal trends was obtained between the Eastern part (avg. $R_s = 0.34$, $p < 0.01$, Stdv = 0.33) and the Western part (avg. $R_s = 0.8$, $p < 0.01$, Stdv = 0.53). The anchorage area of Mogadishu (Somalia) presented the highest rate of increase in NTL during the study period 2012–2020 with ($R_s = 0.94$, $p < 0.01$), probably reflecting its slow recovery from the long civil war. On the South American continent, on

average, there was a downward trend in NTL with ($R_s = -0.18$, $p < 0.01$, Stdv = 0.41). In Northern America, NTL values increased with an average value of $R_s = 0.27$, $p < 0.01$, Stdv = 0.27. The NTL values in the Persian Gulf (Figure 10b) increased similarly to those in North America ($R_s = 0.25$, $p < 0.01$) but with a wider deviation between anchorage areas (Stdv = 0.47), mainly due to a decrease in the Qatar anchorage areas. The NTL values in China (Figure 10c) increased on average by $R_s = 0.32$, $p < 0.01$ and Stdv = 0.32, with only Turkey surpassing China. In Australia, there was a relatively stable neutral trend in NTL values. Among the countries, the decrease in NTL values occurred mainly in the anchorage areas of Japan (nine anchorage areas), Brazil (seven), Chile (seven), India (seven), while in the ports of Venterminals and Guanta (Venezuala), Capetown (South Africa) the most significant decrease occurred within the investigated period with a negative correlation below than $R_s = -0.72$, $p < 0.01$. A significant increase in NTL values also occurred in the anchorage areas of the ports of Huanghua (China) $R_s = 0.91$, $p < 0.01$, Izmir (Turkey) $R_s = 0.88$, $p < 0.01$, Taman (Ukraine) $R_s = 0.86$, $p < 0.01$, Poti (Georgia) $R_s = 0.83$, $p < 0.01$, Lagos (Nigeria) $R_s = 0.82$, $p < 0.01$, and Basrah (Iran) $R_s = 0.78$, $p < 0.01$.

**Table 3.** Matrix of Spearmen correlation coefficients of monthly statistical parameters for the port of Santos (Brazil) and VIIRS monthly values for the period of January 2016–March 2020 (*n* = 51). Positive correlation coefficients greater than 0.4 are highlighted.

| | Remote Sensing | | Official Statistics from the Port of Santos | | | | | | | | | | |
|---|---|---|---|---|---|---|---|---|---|---|---|---|---|
| | VIIRS monthly sum | Ships counted Sentinel 1 | Exports | Imports | Ships waiting | Ships waiting > 72 h | General cargo ships | Bulk solid ships | Tankers | Passenger ships | Roll–on/roll–off ships | Others ships | Total number of ships |
| VIIRS monthly sum | 1 | | | | | | | | | | | | |
| Ships counted Sentinel 1 | 0.51 | 1 | | | | | | | | | | | |
| Exports | 0.37 | 0.59 | 1 | | | | | | | | | | |
| Imports | 0.24 | 0.34 | 0.42 | 1 | | | | | | | | | |
| Ships waiting | 0.17 | 0.44 | 0.82 | 0.45 | 1 | | | | | | | | |
| Ships waiting > 72 h | 0.41 | 0.68 | 0.78 | 0.25 | 0.72 | 1 | | | | | | | |
| General cargo ships | −0.20 | 0.03 | 0.23 | 0.20 | 0.54 | 0.06 | 1 | | | | | | |
| Bulk solid ships | 0.41 | 0.50 | 0.87 | 0.39 | 0.85 | 0.80 | 0.18 | 1 | | | | | |
| Tankers | −0.07 | 0.15 | 0.21 | 0.38 | 0.34 | 0.19 | 0.05 | 0.12 | 1 | | | | |
| Passenger ships | 0.09 | −0.12 | −0.48 | −0.29 | −0.62 | −0.38 | −0.34 | −0.54 | −0.31 | 1 | | | |
| Roll–on/roll–off ships | −0.19 | −0.06 | 0.20 | −0.15 | 0.23 | 0.26 | −0.10 | 0.13 | 0.28 | −0.36 | 1 | | |
| Others ships | 0.03 | 0.03 | −0.10 | 0.29 | −0.11 | −0.13 | 0.05 | −0.14 | 0.05 | 0.12 | −0.16 | 1 | |
| Total number of ships | 0.22 | 0.48 | 0.74 | 0.38 | 0.85 | 0.67 | 0.47 | 0.72 | 0.33 | −0.22 | 0.10 | −0.02 | 1 |

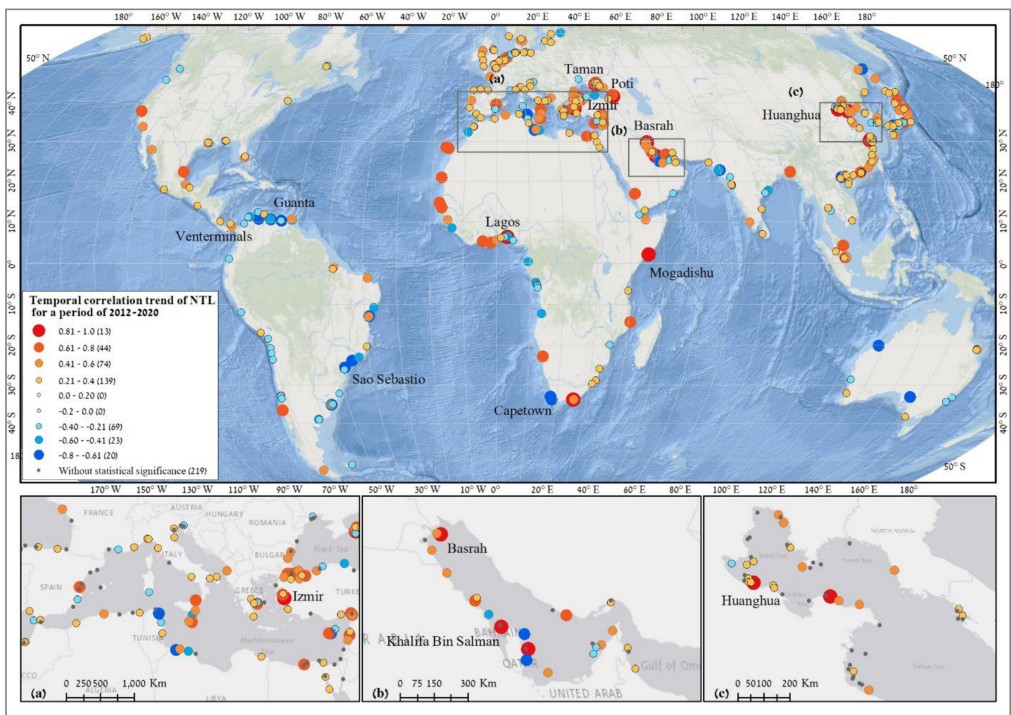

**Figure 10.** Global map of anchorage areas and their temporal trends over the 2012–2020 period. The temporal trends were assessed using the Spearmen correlation. The number in brackets represents the number of anchorage areas in each group of temporal trends. Enlarged maps show the following areas: (**a**) The Mediterranean Sea; (**b**) The Persian Gulf; (**c**) The Yellow Sea region.

### 3.3. Statistical Analysis at the Country–Level

At the country level, we found statistically significant correlations for 10 of the explanatory variables (Figure 11). The annual average values of NTL were very strongly correlated with the CPT ($R_s$ = 0.84, $p < 0.01$), and strongly correlated with the country's population ($R_s$ = 0.68, $p < 0.01$), maximum cargo capacity of the vessels ($R_s$ = 0.66, $p < 0.01$), average import of the country ($R_s$ = 0.62, $p < 0.01$), GDP ($R_s$ = 0.61, $p < 0.01$), and LSCI ($R_s$ = 0.6, $p < 0.01$). Moderate correlations were found for maximum container carrying capacity ($R_s$ = 0.55, $p < 0.01$), maximum vessel size ($R_s$ = 0.51, $p < 0.01$), and port calls $R_s$ = (0.42, $p < 0.01$).

Figure 12 provide a scatter plot of the correlation between CPT and NTL values at the country level. As shown in Figure 11, NTL data were most strongly and significantly correlated with "CPT" at the state level ($R_s$ = 0.84, $p < 0.01$). China ranked first among the countries in terms of CPT (223,809,105 TEU), was four times higher than the USA CPT (52,716,134 TEU), and its average annual SOL was also four times higher (3876 and 943 TEU).

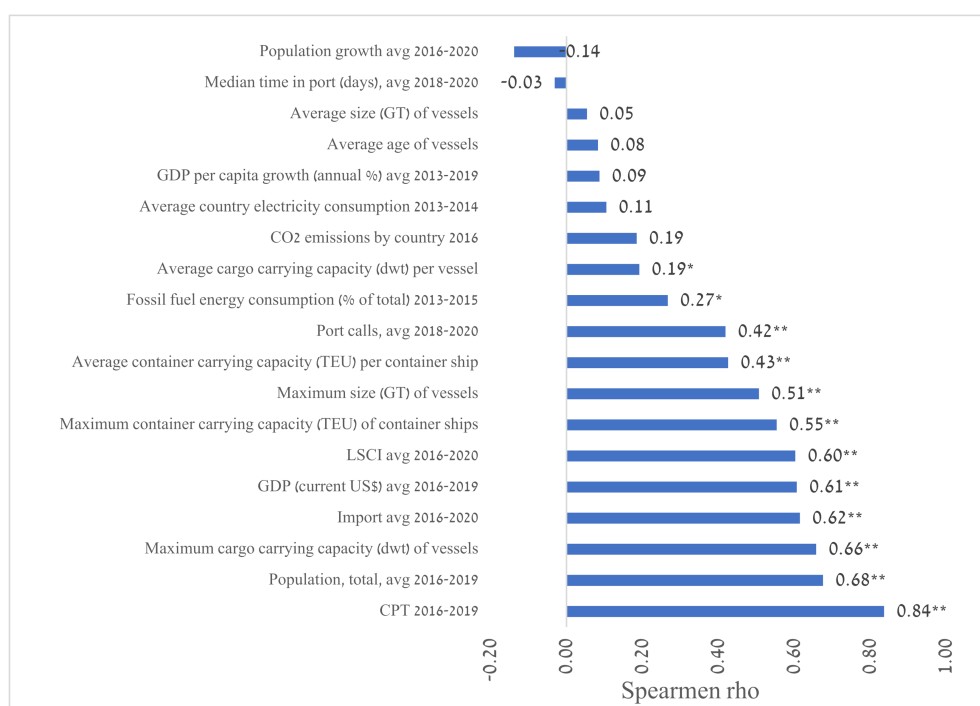

**Figure 11.** Spearmen correlation coefficients at the country level (* $p < 0.05$, ** $p < 0.01$; $n = 97$) for the response variable of average annual SOL values. Variables are ordered by the magnitude of their correlation coefficient with the SOL at the country level.

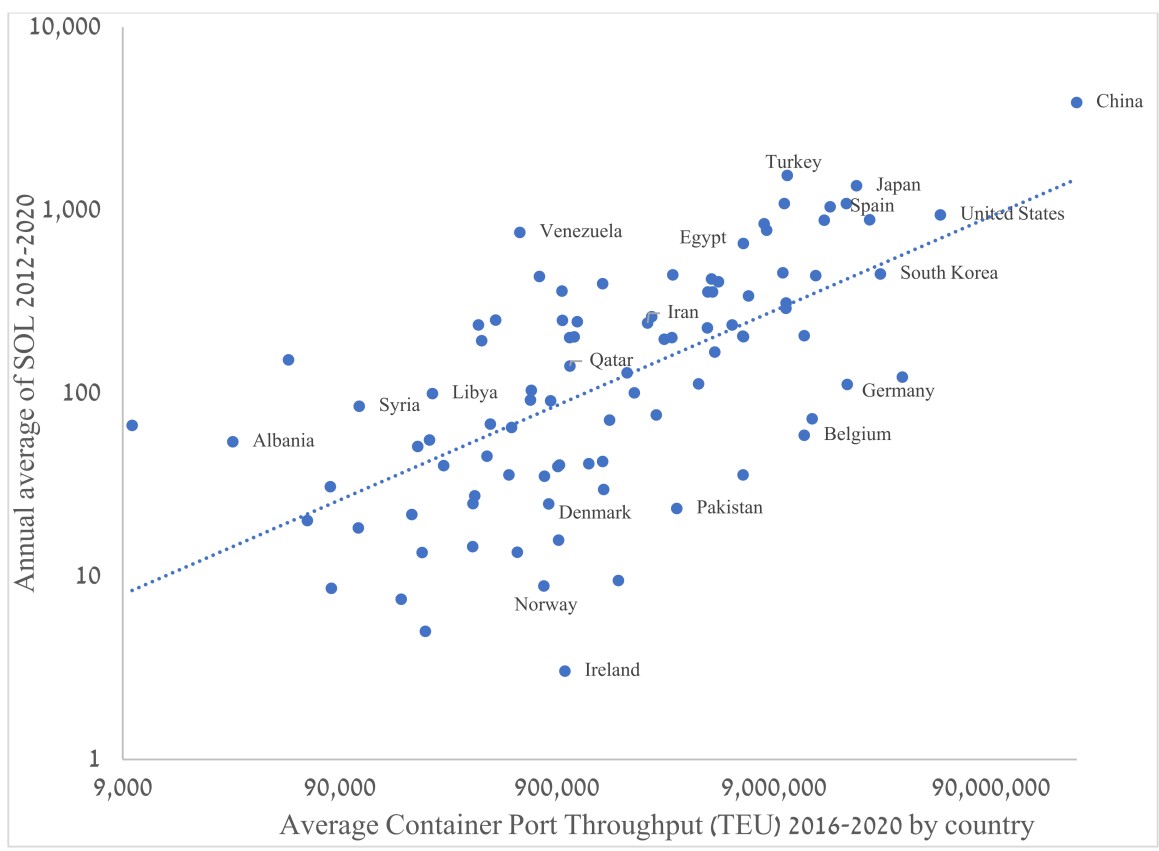

**Figure 12.** Scatter diagram of the correspondence ($R_s = 0.84$, $p < 0.01$) between the average Container Port Throughput (CPT) 2016–2020 by country and averages annual SOL for the period of 2012–2020, N = 97.

## 4. Discussion

To track global and local traffic flows and assess the state of development of the maritime industry, accurate data on port shipping activities is essential. Information of this nature is crucial for managers and analysts to make strategic decisions and monitor the maritime industry's progress towards its management goals.

Sources of artificial lights in the marine realm vary and include coastal cities, oil rigs, harbor lighting, fisheries, and anchoring ships [50]. The use of night–time remote sensing data for ship surveillance is a relatively new research field. Night light remote sensing technology has a unique night vision capability that can overcome the problem that traditional optical daytime remote sensing images cannot track targets at night. In existing studies, the VIIRS data on night illumination is widely used in the socio–economic sphere [28,51,52], the development of algorithms and applications for monitoring fishing vessels [19,53,54], assessment of the airport's throughput of traffic flow [51], assessment port economics scores [55], as well as assessment of countries' economic situation during the crisis [56]. Using VIIRS night light data, we showed that it is possible to assess shipping activities at the port level for which there is little or no consistent information at the global level. The NTL data on port anchorage areas are therefore a vital source of data for calibration and validation of, for example, the port collaborative decision making (PortCDM) concept [57], UNCTAD maritime and shipping parameters [33,58], and country economic assessments. Shipping activity in the water area near the port (anchorage area) proceeds without respite around the clock. To ensure normal operation at night–time, similarly to daytime, electricity is used on the ship, including for lighting the ship. This study is based on large data sets (601 anchorage areas) covering most of the anchorage areas of the world's major ports over the period from April 2012 to March 2020, thus providing a method for estimating both the average monthly number of ships in the anchorage area and various shipping and trade variables. Eight of the explanatory variables presented in Table 2 were significantly correlated ($R_s > 0.50$, $p < 0.01$) with the annual average (2012–2020) SOL values of VIIRS at the country level. The CPT index, which provides information on the number of unloaded and loaded containers by the port, was strongly correlated ($R_s = 0.84$, $p < 0.01$) with the NTL values. A possible reason might be that in most of the ports there is infrastructure for loading and unloading containers from ships as well as from container trucks (IMO, 2021). Moreover, based on other independent UNCTAD parameters (Table 2), a high correlation was obtained for the annual shipping parameters related to the aspects of "maximum" (instead of average): the maximum cargo carrying capacity (dwt) of vessels ($R_s = 0.66$, $p < 0.01$), the maximum container carrying capacity (TEU) of container ships ($R_s = 0.55$, $p < 0.01$), and the maximum size (GT) of vessels ($R_s = 0.51$, $p < 0.01$). A similar assessment of the use of night–time lighting data as an indicator has also been provided in studies of evaluating port economics comprehensive scores (PECS), based on UNCTAD, 1987 [59], with a correlation of ($R^2 > 0.85$) in the case of Shanghai port [55] and even an assessment of airport throughput (represent the annual number of aircraft movements or passengers [60]) with a significant correlation of ($R^2 > 0.85$) [51]. The domination of "Maximum metrics" and their higher correlation with NTL could be the result of port prioritization and commercial considerations by operating companies [61] in the entrance of large ships carrying high amounts of cargo [62]. Such prioritization leads to the long waiting time of smaller ships that wait for their queue to enter the port in the anchorage area [63,64]. Moreover, we also found significant correlations of NTL with countries' socio–economic parameters such as: country's average population 2016–2020 ($R_s = 0.68$, $p < 0.01$), average import 2016–2020 ($R_s = 0.62$, $p < 0.01$), and average GDP ($R_s = 0.61$, $p < 0.01$) [65], despite the relative decline in the importance of cities with large populations in global traffic [66]. Countries with large populations and high GDP often require high levels of imports and a large number of different ship segments that serve a high standard of living a large country's population. This conclusion is supported by the strong correlation of the LCSI parameter with NTL ($R_s = 0.60$, $p < 0.05$), which indicates the variability level of the country's integration into the global liner transportation networks [33,58]. The use of fossil

fuels for energy ($R_s$ = 0.27, $p < 0.05$) and $CO_2$ emission ($R_s$ = 0.19) were not found to be strongly correlated with NTL, probably because the ships are not a significant factor [67] or are not counted in the overall balance of the country for each of these parameters. Thus, NTL data is an indicator of shipping that can be used to accurately estimate a wide range of activity (port, maritime, country, economy) parameters of ports and country, which is also confirmed by the results Liu (2019), where the researchers show that NTL data is a proxy indicator of economic assessment ports of China, despite high variability of port pixels and light interference around the port area [68]. The results obtained are of applied importance for assessing the dynamics of the development of seaports and the socio–economic parameters of the country, similar to how Bennet, 2017 [69] shows the correlation of NTL with socio–economic parameters on various spatio–temporal scales, thereby compensating for the lack of statistical data on the activities of ports and country parameters. On a local level, the observed negative temporal trend in the Venezuelan anchorage areas (five out of six anchorage areas having a temporal decrease of $R_s < -0.3$, and three with a decrease of $R_s < -0.6$) is confirmed by the results of a study conducted by Zhang et al. 2020 [56]; they assessed the economic crisis in Venezuela using NTL data for April 2012–December 2018, based on 12 cities, finding high correlations ($R^2 > 0.8$) between the sum of urban lights and several economic parameters (crude oil production, USD exchange rate and the number of asylum seekers), thus demonstrating the use of NTL data as an indicator of the economic state of Venezuela during the crisis. At the port level, results of statistical analysis showed lower correlations than at the country level, but the results of a detailed analysis we conducted for the Port of Santos, Brazil, (Table 3) shed light on some of the correspondence between NTL and the number of ships counted in Sentinel 1. In 2019, when we observed a discrepancy between the NTL value and the number of ships counted on Sentinel 1 images, only the parameter "number of ships waiting for more than 72 h" showed a similar trend. The bulk ship segment constitutes the majority of those waiting in the anchorage area of the Santos port ($R_s$ = 0.85) and those that are over 72 h ($R_s$ = 0.67), which are mainly dedicated to exports ($R_s$ = 0.87). Hence, when there is a decrease in exports via bulk ships, fewer ships will be waiting in the anchoring area. Perhaps this is the reason for the discrepancy between the NTL value and the number of ships counted on Sentinel 1 flights in 2019. Although Sentinel 1 SAR data is not affected by cloud cover (whereas cloud cover hampers the detection of night lights by VIIRS), VIIRS acquires night–time imagery every night, whereas the revisit time of Sentinel 1 is lower (six days; [70]). Hence, the two sensors did not acquire their images at the same dates and time of day, which may explain some of the discrepancies between the data from these two sensors. Therefore, the proposed method using night lights as an indicator of ship activity is particularly suitable for assessing spatial and temporal trends in the maritime industry, complementing other methods of tracking ships (AIS, SAR images), especially where official statistics are not available.

A detailed analysis of the causes of temporal variations makes it possible to improve the sensitivity of DNB to changes in illumination [68]. In this study, we developed and implemented several correction methods for VIIRS data to better analyze the light emitted from anchorage areas, although VIIRS significantly improved quality over DMSP/OLS in terms of spatial resolution, dynamic range, quantization, calibrations, and spectral range availability over DMSP–OLS [25]. First, to minimize the influence of temporal variation of natural light such as airglow, we used the method proposed by Coesfeld, 2020 [22]. Thus, fluctuations in natural light sources that limit the ability of night light sensors to detect changes in small artificial light sources have been minimized, increasing the ability to analyze the light emitted by ships. Moreover, city lights also represent an important factor that influences scattering over the marine environment, as light from brightly lit coastal cities can reach considerable distances at sea (Figure 2) by scattering through the atmosphere [24]. As of today, there is no method of amendment developed to minimize the influence of city lights over adjacent coastal waters (to assess the amount of NTL emitted from the coastal waters themselves), and thus we focused on anchorage areas that were too

close to the coast. Secondly, due to the high cloudiness in some areas of the earth, monthly data from VIIRS are often underestimated. To fill in the gaps, we applied an interpolation method for calculating the monthly values for the underestimated months based on the VIIRS values of 12 adjacent months. Results of the *t*-test showed that the interpolation method filled the data in cloudy months and did not affect the original data.

The use of VIIRS data has certain limitations. Among the limitations is the coarse spatial resolution of 742 m [25], which does not allow a more detailed analysis of small dense areas and objects. Cloudiness, which is in principle a frequent occurrence near coastal areas, and in some regions (e.g., the tropics) a frequent occurrence, does not allow information to be collected on a large percentage of the days of the year. Moreover, the coastal light emitted by cities and ports themselves is much stronger than the light emitted by ships, which in some cases makes this approach ineffective [71] if an anchorage area is located too close to a brightly lit city. Additional sources of variation in emissions of NTL from ships are associated with the types of ships in anchorage areas and their night lights used. In this study, we did not have data on ship types (e.g., oil tankers, cargo ships, etc.), which lowered our ability to explain the variability in NTL between anchorage areas. AIS data can be a suitable source for replenishing knowledge about the types of ships in future studies. In the case of comparison between the anchorage areas, there may be variations in the volumes and powers of lighting permitted by local port authorities. Moreover, this difference can also exist between the ports of the same country, and even the policy of the port about the use of night light can change over time, which can lead to temporal changes. The VIIRS sensor is panchromatic and does not measure night light in the blue channel, thereby losing the night light emitted by ships in blue wavelength (which is a significant component of light emitted by LED lightings) [72]. Finally, due to the wide coverage on the ground by a single image, VIIRS imagery is mostly not acquired at nadir [22,73], and changes in the zenith and azimuth view angles may affect the amount of light received by the sensor from the ships, as was documented for light emissions from cities [74].

The ability of NTL data to serve as a proxy for shipping activity also depends on the number of anchorage points and their density, the type of ships entering the port, and waiting times in the anchorage area. The world's standards for the construction of ships as well as the rules on movement at sea, require ships to be equipped with a variety of lights, most of which are standardized [75,76]. Consequently, more sensitive sensors will enable measurement and distinguish the NTL values for different types of shipping segments [54]. For example, fishing vessels are the most illuminated ships at night–time. Depending on the type of fishing, fishery vessels have different types and directions of night–time lights, both for fishing and for working onboard [77]. Generally, fishing is prohibited in the harbor area and at most anchorages; therefore, their night–time light should not affect the light emitted from anchorage areas. Moreover, in ports with long queues, a ship that waits for more than one night will increase its night signature compared with ships or anchorage areas with short anchorage time (without anchorage during night–time hours). All these factors affect the ability of satellite sensors to capture night light produced by ships and to be used as an indicator for shipping activity. For example, the ports of Fujairah and Malta are mostly hosting tankers [78]. Their anchorage areas are located a few kilometers from the city coast and the light they emit, which makes it possible to measure mainly the light emitted by one sector of the maritime industry, while such an assessment is almost impossible inside the port due to the strong light emitted by the port infrastructure [68]. Thus, for remote ports that are not exposed to coastal lights, it is maybe easier to use night light values as a proxy for shipping activities, considering additional parameters such as ship size, anchorage point density, etc. In ports with a small number of anchorage points and with a small number of vessels in the anchorage area, the proposed method of using night lights may be limited in its ability to assess shipping activities, as not enough lights will be emitted that can be captured by the VIIRS sensor, as also found for other small scale economic activities [79].

## 5. Conclusions

In this study, we carried out a global assessment of shipping activity using the VIIRS satellite data by measuring the night lights emitted from ships at anchorage areas. The analysis was carried out at three geographic levels: overtime at the anchorage level using the example of the port of Santos (Brazil), overall anchorage areas at the country level ($n = 97$), and across all anchorage areas globally ($n = 601$).

The main conclusion of this study is that monthly/annual VIIRS data can serve as a good proxy for estimating the number of vessels as well as various shipping metrics (such as CPT, LSCI) in anchorage areas at the port and country levels. The estimation of the number of ships in anchorage areas with a small number of ships is probably limited due to the low energy of night light emitted by a small number of ships, and in such cases, VIIRS data cannot be used as an indicator. VIIRS NTL data can be implemented in a wide global range of studies of shipping and for assessing the economic development of ports and country parameters. Moreover, this method allows analyzing shipping, ports, and countries parameters, for which we have obtained a significant correlation with NTL data, for example, container port throughput. As a result, we conclude that NTL data can be used as an indicator for a wide range of assessments of ports, countries, and the shipping industry in general, and is applicable for ports and countries that do not share information, as well as for tracking spatial and temporal trends.

Finally, the results should be useful to international maritime organizations, governments, policy–makers, and stakeholders in formulating effective strategies for developing tools to assess shipping activities in the anchorage area and their use in overall port operations.

**Author Contributions:** Conceptualization S.P., N.L. and R.B.; methodology S.P., N.L. and R.B.; software S.P. and N.L.; validation S.P. and N.L.; formal analysis S.P. and N.L.; investigation S.P., N.L. and R.B.; resources S.P. and N.L.; data curation S.P. and N.L.; writing—original draft preparation S.P.; writing—review and editing S.P., N.L. and R.B.; visualization S.P. and N.L.; supervision N.L. and R.B.; project administration S.P., N.L. and R.B.; funding acquisition S.P. All authors have read and agreed to the published version of the manuscript.

**Funding:** S.P. is grateful to the Department of Marine Geoscience, the Maritime Policy and Strategy Research Center (HMS), and the Wydra Institute of Shipping and Aviation Research of the University of Haifa in Israel for their financial support.

**Conflicts of Interest:** The authors declare no conflict of interest.

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
