# Peer review of "A Global Assessment of Night Lights as an Indicator for Shipping Activity in Anchorage Areas"

_remotesensing, doi:10.3390/rs14051079_

Round 1

Reviewer 1 Report

This paper presents a study on indicating global shipping activity in anchorage areas using NPP/VIIRS nighttime light data. It is a very interesting study but the writing, analysis and figures are a little rough. I think that it needs significant improvements to be worth to publish in RS. Main comments: - The first two paragraphs in Section 1.3 should be integrated into the first paragraph of Section 1 to well illustrate the significance of this topic. - Section 1.1 and 1.2 have little relationship with this paper and should be deleted. - The analysis at the country level is of little significance. I suggest focusing on the port level and carry out in-depth and comprehensive analysis. - The authors shall give some detailed maps of several typical areas for specific analysis, such as China coast, the Persian Gulf and the Mediterranean Sea. - From Fig. 8, there are large difference between the vessel number derived from Sentinel 1 data and the average NTL from NPP/VIIRS data. Why? An in-depth analysis is needed. - The temporal variations of several typical areas should be given and analyzed. - The discussion section is really weak. Some contents in this section that are not closely related with this study should be deleted. The results of this study should be compared with other studies and the intellectual merits of this paper should also be discussed in this section. Other comments: - P3, L107. The definition of the anchorage area should be moved into the beginning of the Introduction Section. - P4, Fig. 1. A legend is needed. - P5, It seems that the spatial scopes of the two images are not the same, which can be judged by the shapes and locations of the costal lines. What do the two red boxes represent in the right map? In addition, it is better to use a Sentinel 1 image acquired in 2019 for comparison. - P5, L153. The spatial resolution of 0.004167 degree is not accurate because the distance of 1 degree varies with latitude. - P9, L277. Where is this Fig. 4? - P9, Fig. 4. The scale bar seems wrong. - P10, Fig. 5. I suggest using different color to indicate the SOL for better visual effects and better discrimination with Fig. 4. - P11, Fig. 6. The figure should be redrawn to improve the quality. The correlation coefficient and the sample number can be labeled in the figure. And more detailed analysis on Fig. 6 is needed.

Author Response

We would like to thank the Editor and the two reviewers for their constructive comments and suggestions.

Reviewer 2 Report

The manuscript presents the enormity of the analyzes performed. However, I am asking the authors for a better explanation of the need for such analyzes (what is their advantage over classical methods of vessel traffic monitoring). Because apart from the enormous amount of work, I do not see the significance and the possibility of using the final results. We only have data on the number of vessels. As the analyzes have shown, we do not have additional information, e.g. on the size of such a ship. I would also like the authors to define the limitations of research more broadly (maybe a separate subchapter).

Additional remarks: Section 1.2. Please focus more on comparing the limitations and possibilities of using this type of data for the analysis covered in the work (better overview of sources on the technical aspects of the data used, e.g. hours of data collection). You can also do this in section 2.2. Figure 3 Vertical axis (2 - square value), horizontal axis (add month data), is the radiance value the average for the indicated area - or is the data from a specific 1 pixel?

Author Response

(The authors gave the same response as above.)

Round 2

Reviewer 1 Report

The revision made significant improvement according to the comments. There are just two important points to be modified or addressed:

- The analysis at the country level is of little significance. I suggest focusing on the port level.

- From Fig. 9, there are large differences between the vessel number derived from Sentinel 1 data and the average NTL from NPP/VIIRS data. Why? An in-depth analysis is needed.

BTW, the authors shall give a point-to-point response.

Author Response

Dear Dr. Ran Goldblatt, Dr. Steven Louis Rubinyi, Dr. Hogeun Park

We would like to thank the Editors and the reviewer for the constructive comments and suggestions in this second review.

Following are our replies (in blue) to the comments in the second round of reviews (“minor revision”).

We hope that our manuscript can now be accepted.

Kind regards,

Semion Polinov, Revital Bookman and Noam Levin
